# Evaluation of Computer Vision Systems and Applications to Estimate Trunk Cross-Sectional Area, Flower Cluster Number, Thinning Efficacy and Yield of Apple

**Luis Gonzalez Nieto [1,\*]**, **Anna Wallis [2]**, **Jon Clements [3]**, **Mario Miranda Sazo [4]**, **Craig Kahlke [4]**, **Thomas M. Kon [5]** and **Terence L. Robinson [1]**

[1] Horticulture Section, School of Integrative Plant Science, Cornell University, Geneva, NY 14456, USA; tlr1@cornell.edu

[2] Extension Section, Michigan State University, Grand Rapids, MI 49503, USA; aew232@cornell.edu

[3] UMass Extension Section, University of Massachusetts, Belchertown, MA 01007, USA; jon.clements@umass.edu

[4] Cornell Cooperative Extension, Lake Ontario Fruit Program, Albion, NY 14411, USA; mrm67@cornell.edu (M.M.S.); cjk37@cornell.edu (C.K.)

[5] Mountain Horticultural Crops Research and Extension Center, Department of Horticultural Sciences, North Carolina State University, Mills River, NC 28759, USA; tom_kon@ncsu.edu

[\*] Correspondence: lg579@cornell.edu

**Abstract:** Precision crop load management of apple requires counting fruiting structures at various times during the year to guide management decisions. The objective of the current study was to evaluate the accuracy of and compare different commercial computer vision systems and computer applications to estimate trunk cross-sectional area (TCSA), flower cluster number, thinning efficacy, and yield estimation. These studies evaluated two companies that offer different vision systems in a series of trials across 23 orchards in four states. Orchard Robotics uses a proprietary camera system, and Pometa (previously Farm Vision) uses a cell phone camera system. The cultivars used in the trials were 'NY1', 'NY2', 'Empire', 'Granny Smith', 'Gala', 'Fuji', and 'Honeycrisp'. TCSA and flowering were evaluated with the Orchard Robotics camera in full rows. Flowering, fruit set, and yield estimation were evaluated with Pometa. Both systems were compared with manual measurements. Our results showed a positive linear correlation between the TCSA with the Orchard Robotics vision system and manual measurements, but the vision system underestimated the TCSA in comparison with the manual measurements ($R^2$s between 0.5 and 0.79). Both vision systems showed a positive linear correlation between nubers of flowers and manual counts ($R^2$s between 0.5 and 0.95). Thinning efficacy predictions (in June) were evaluated using the fruit growth rate model, by comparing manual measurements and the MaluSim computer app with the computer vision system of Pometa. Both systems showed accurate predictions when the numbers of fruits at harvest were lower than 200 fruit/tree, but our results suggest that, when the numbers of fruits at harvest were higher than 200 fruit/tree, both methods overestimated final fruit numbers per tree when compared with final fruit numbers at harvest ($R^2$s 0.67 with both systems). Yield estimation was evaluated just before harvest (August) with the Pometa system. Yield estimation was accurate when fruit numbers were fewer than 75 fruit per tree, but, when the numbers of fruit at harvest were higher than 75 fruit per tree, the Pometa vision system underestimated the final yield ($R^2 = 0.67$). Our results concluded that the Pometa system using a smartphone offered advantages such as low cost, quick access, simple operation, and accurate precision. The Orchard Robotics vision system with an advanced camera system provided more detailed and accurate information in terms of geo-referenced information for individual trees. Both vision systems evaluated are still in early development and have the potential to provide important information for orchard managers to improve crop load management decisions.

**Keywords:** computer vision system; trunk cross-sectional area (TCSA); bloom intensity; fruit growth rate model; yield estimation





## 1. Introduction

Precision crop load management is the single most important yet difficult management strategy that determines the annual profitability of apple orchards [1]. Apple trees generally produce too many flower clusters and fruit for an optimum crop load [2]. Only about 3–10% of the initial fruit population should be carried to harvest to optimize crop value and promote annual bearing. When the fruit set is too high, low-quality fruit are produced and biennial bearing is induced [3]. There is a positive correlation between crop load and high fruit yield [4]. Crop load is a common parameter used in the industry that and can be defined simply as the number of fruits per tree or, more commonly, as the number of fruits per trunk cross-sectional area (fruit/TCSA) [5–7]. TCSA is often used to define the optimum target crop load for a tree or an orchard. Several studies have estimated that the optimal crop load in apple is between six and eight fruits/TCSA [1,8,9]. The traditional method of calculating TCSA is to measure the trunk circumference at 30 cm above the graft union with a tape measure or caliper and then calculate the cross-sectional area [10]. Recently, several studies have focused on using computer vision systems to estimate the TCSA [8,11,12].

The concepts of precision crop load management were developed to help growers manage crop load in a systematic manner to optimize fruit load and crop value [13]. The process involves precision pruning, precision chemical thinning, and precision hand thinning. Precision pruning is a process of the stepwise removal of flower buds to a pre-determined flower bud load before bud break in the spring. Precision chemical thinning is the sequential application of chemical thinning sprays, guided using computer models to adjust the rate and timing of chemical application. The models include the pollen tube growth model (PTGM) to time blossom thinning spray applications precisely [14–17]. This model requires a visual observation of the percentage of open blossoms. However, visual observation of open blooms is a subjective method.

The other two predictive models, the MaluSim model [18] and the fruit growth model, use precision crop load management [19]. The MaluSim model is a dynamic simulation model that uses weather variables to predict apple tree carbohydrate supply and demand [18], and is used to predict the rate of application and thinning efficacy. The fruit growth rate model is used to predict thinning efficacy after thinner application and determine whether additional applications are needed. It assumes faster growing fruitlets will persist and slower growing fruitlets will fall off. This model has been very useful in precision crop load management but its adoption has been limited due to the time requirement to carry out repeated manual measurements of fruitlet diameters.

Harvest time is another critical time when information estimates of fruit numbers and size are very valuable. Accurate forecasts of yield and fruit size can inform decisions for harvest labor needs and marketing plans for fruit growers and packing companies [20]. Presently, manual counting or visual assessments are used. These strategies can be difficult to execute during the harvest period and are imprecise.

Recent advancements in digital technology and computer vision are means to automate several steps in precision crop load management to manage pruning, flower and fruit thinning, and harvest more efficiently. The most important steps are automating the counting of flower buds in the dormant season, counting of flower clusters during bloom, measuring fruitlet diameters during the post-bloom thinning period, and counting final fruit numbers and measuring fruit size before harvest. This information is needed to inform management decisions [21]. Over the last several years, research has focused on evaluating several digital ag companies on computer vision tools to streamline the counting of apple buds, flowers, fruitlets, and fruit at harvest to guide human or robotic workers to optimize crop load while minimizing labor costs [22]. The objective of the current study was to evaluate the accuracy of several commercial computer vision systems and computer applications to estimate trunk cross-sectional area (TCSA), flower cluster number, thinning efficacy, and yield estimation.

## 2. Materials and Methods

### 2.1. Plant Material and Sites

Experiments were conducted in 2021, 2022, and 2023 in commercial apple (*Malus domestica* Borkh.) orchards in Massachusetts, Michigan, New York, and North Carolina (Figure 1 and Tables 1–4). Mature and uniform apple trees at each location were used in all trials with one exception. Specifically, 3-year-old 'Granny Smith' apples were evaluated in one experiment in North Carolina (Tables 1–4). The cultivars used in the trials were 'NY1', 'NY2', 'Empire', 'Granny Smith', 'Gala', 'Fuji', and 'Honeycrisp' (Tables 1–4). Orchards were managed with standard commercial practices used in each region.

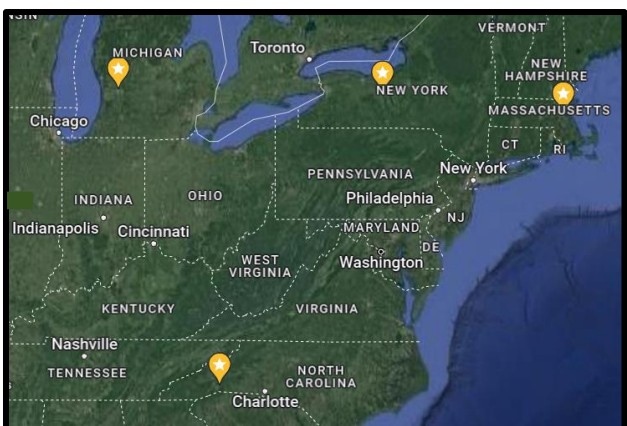

**Figure 1.** Locations in the Eastern USA of the experiments conducted on digital vision systems in 2021, 2022, and 2023. Geolocation Massachusetts (42°15′13.3″ N 72°21′35.3″ W), Michigan (42°58′15.1″ N 85°40′17.6″ W), New York (42°52′39.9″ N 77°00′25.9″ W) and North Carolina (35°25′47.9″ N 82°33′29.9″ W).

### 2.2. Trunk Cross-Sectional Area (TCSA)

The evaluations of TCSA were conducted in 2023 with the Orchard Robotics vison system and manual measurements in an orchard in Geneva, New York. Full rows of 'Gala' (1–90 trees, 2–30 trees), 'Honeycrisp' (90 trees), and 'Fuji' (30 trees) were evaluated (Table 1 and Figure 2). A tape measure was used for manual measurements of trunk circumference and converted to square centimeters (cm$^2$) cross-sectional areas. The TCSA was measured at 30 cm above the graft union. The Orchard Robotics vision system takes a photo of each single tree and detects the visible portion of the trunk (between graft union and first branches), and then measures the trunk diameter in the middle of the section that it can detect and calculates the TCSA.

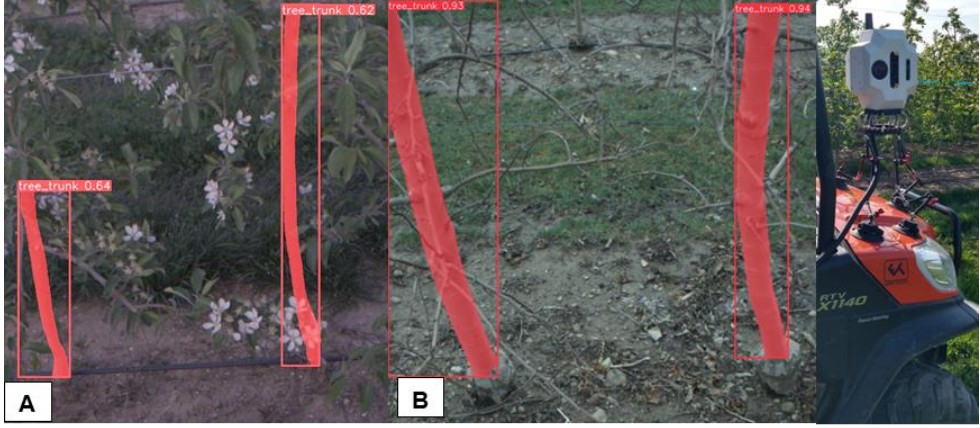

**Figure 2.** Trunk detections and measurements of TCSA with the Orchard Robotics vision system. Picture (**A**) is of two young trees (5 years old) and picture (**B**) is of two older trees (17 years old).

**Table 1.** Characteristics of the orchards used for TCSA estimates in 2023.

| Location | Cultivar | Year Planted | Rootstock | System | Spacing (m) (Number Tree per Row) |
|----------|----------|--------------|-----------|--------|-----------------------------------|
| New York | 'Gala'-1 | 2006 | G.11 | Tall Spindle | 0.9 × 3.4 (97) |
| | 'Honeycrisp' | 2006 | M.9 | Tall Spindle | 0.9 × 3.4 (97) |
| | 'Gala'-2 | 2019 | G.41 | Tall Spindle | 0.9 × 3.4 (30) |
| | 'Fuji' | 2019 | G.41 | Tall Spindle | 0.9 × 3.4 (30) |

*2.3. Blossom Counts*

In 2023, the accuracy of two computer vision systems was evaluated in New York (Table 2). Manual and computer vision estimates of the number of flower clusters per tree were carried out at the pink-bud stage (BBCH 61-65). The Pometa vision system was evaluated in a commercial orchard on 'Gala' and 'NY1' (Table 2). Pometa cluster counting was a 2023 beta product. Five representative trees were selected per block, which were scanned using a video function on a compatible cellular phone (Apple iPhone 14 pro (Cupertino, CA 95014, USA)), according to the company's directions (Figure 3). Two videos were taken of each tree (one on each side), with a total of ten videos. Videos were uploaded to the Pometa website for data processing. After 24 h, results of the processed videos were available and were downloaded.

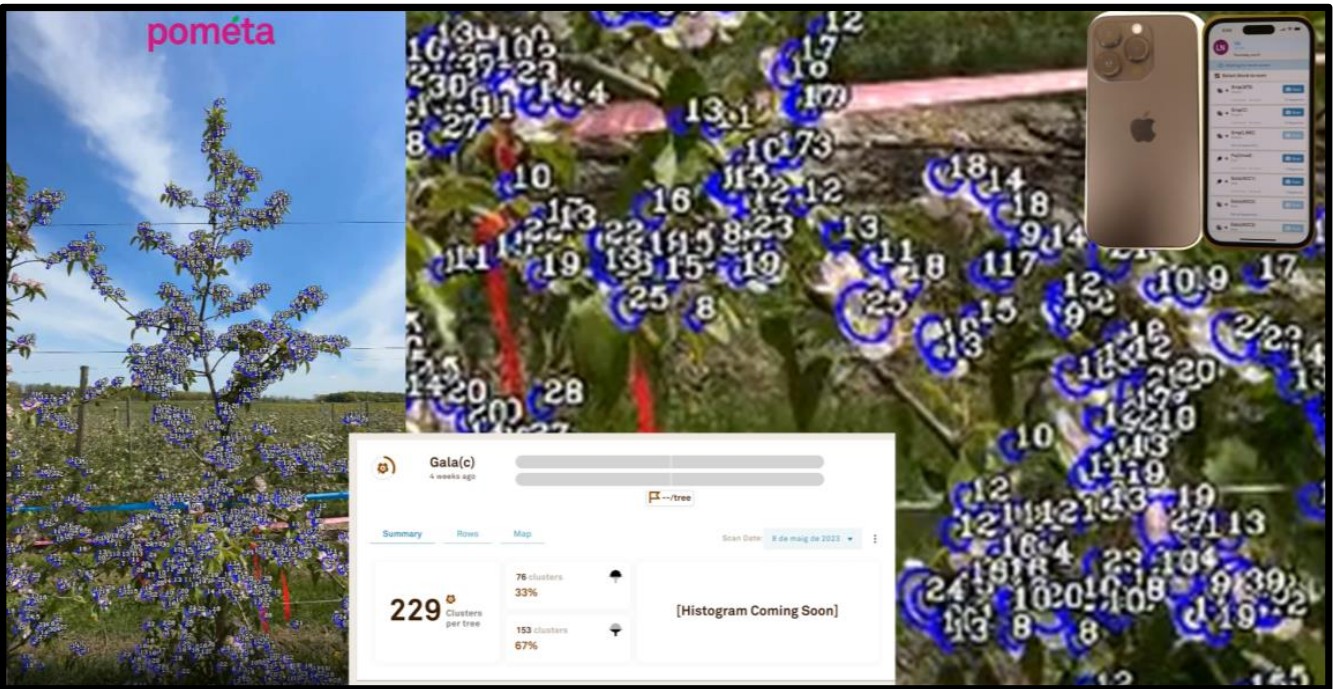

**Figure 3.** Flower clusters identified with the Pometa vision system and information derived from the cluster counts. Blue circles illustrate where flowers were detected.

The Orchard Robotics vision system was evaluated in a commercial orchard of 'Gala' and 'Fuji' (Table 2). One full row (130 trees) for each variety was scanned with the Orchard Robotics system, while 30 trees in each row were selected for manual counts. The Orchard Robotics vision system was evaluated using manufacturer recommendations. At full bloom, one side of the row (well-exposed, sunny side) was scanned with the camera system mounted on an orchard vehicle at 8 km/h. All scans were uploaded automatically from the cam to the Orchard Robotics website for processing (Figure 4). After 24 h, the results of the processed videos were available on the company's website. However, it was necessary to

record the manual counts at the same time as the evaluation of the representative 5 trees to calibrate the system.

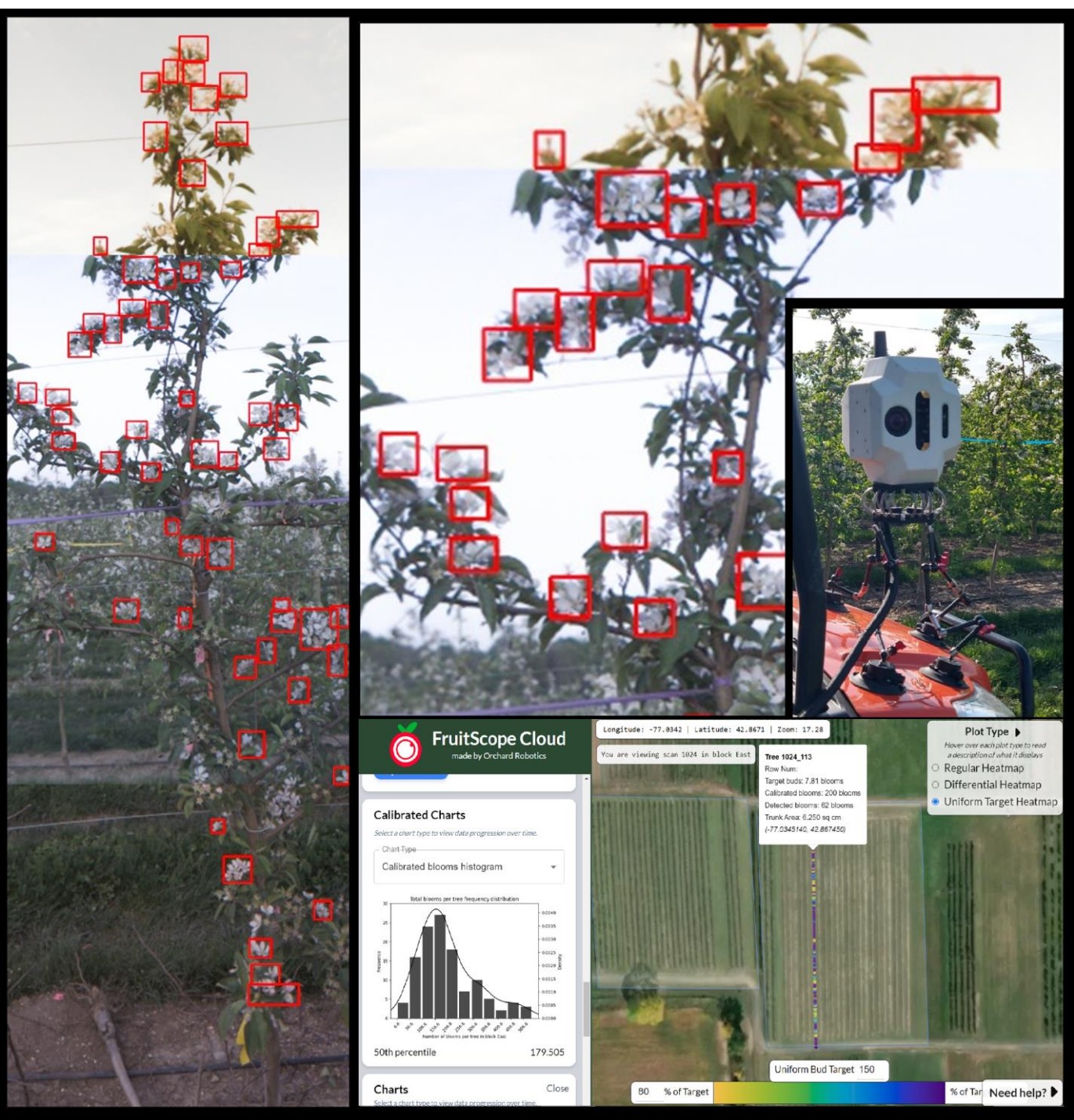

**Figure 4.** Flower clusters identified with the Orchard Robotics vision system and information derived from the cluster counts. Red squares illustrate where flower clusters were detected.

**Table 2.** Characteristics of the orchards used to evaluate bloom intensity in 2023.

| Location | Cultivar | Rootstock | System | Spacing (m) | 2023 | |
| --- | --- | --- | --- | --- | --- | --- |
| | | | | | Pometa (n of Trees) | Orchard Robotics (n of Trees) |
| New York | 'NY1' | G.41 | Tall Spindle | 0.9 × 3.4 | 4 | |
| | 'Gala' | G.41 | Tall Spindle | 0.9 × 3.4 | 6 | 30 |
| | 'Fuji' | G.41 | Tall Spindle | 0.9 × 3.4 | | 30 |

### *2.4. Thinning Efficacy*

The trials were carried out in 13 commercial orchards in 2022 in New York, Massachusetts, Michigan, and North Carolina. 'Gala', 'Fuji', and 'Honeycrisp' cultivars were evaluated with Pometa scans and with an iPhone, and compared with the MaluSim app (manual measurements) (Table 3). At each orchard, five representative trees were selected and the total number of blossom clusters per each tree were counted. The final number of fruits was determined after natural fruit drop or at harvest. Commercial chemical thinning sprays were applied in each orchard between 6 and 8 mm fruit king diameter.

The MaluSim app was developed by Cornell University and includes the fruitlet growth rate model, apple carbohydrate model, and an irrigation model. The MaluSim fruit growth model is based in the traditional fruit growth model developed by Greene et al. [19]. The model began with the selection of five representative trees and the tagging of 15 clusters in each tree (75 clusters in total). After the thinning spray was applied, the fruit diameters of all fruitlets on the tagged cluster were measured at days 4 and 7 after application with a digital caliper. The total number of flower clusters per tree, number of flowers per cluster, target of fruit per tree, and all diameter measurements were entered into the MaluSim app, which then predicted the thinning efficacy and average number of fruits per tree (Figure 5).

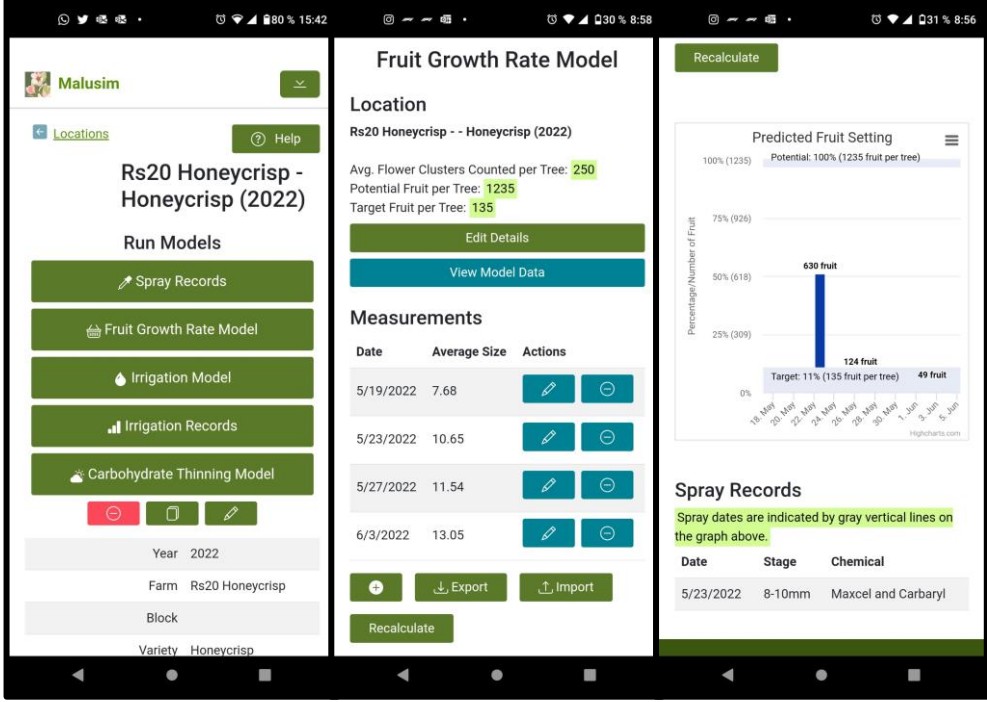

**Figure 5.** Flower set estimation with the MaluSim app at one location in New York in 2022.

The Pometa system of predicting thinning efficacy after a thinning spray is done by scanning the trees using an iPhone video recording of the tree at days 4 and 7 after thinner application. The same trees used for the MaluSim estimation of thinning efficacy were 'used for the Pometa thinning efficacy estimation. The Pometa scanning was done using manufacturer recommendations and equipment: (smart phone with a stereo video camera, and enhanced GPS location identifier) (Figure 6). Two videos were captured of each tree (one video for each side of the tree and a total of ten videos). The videos were uploaded to the Pometa web site for processing. After 24 h the results of the processed videos were available at the company's website (Figure 6).

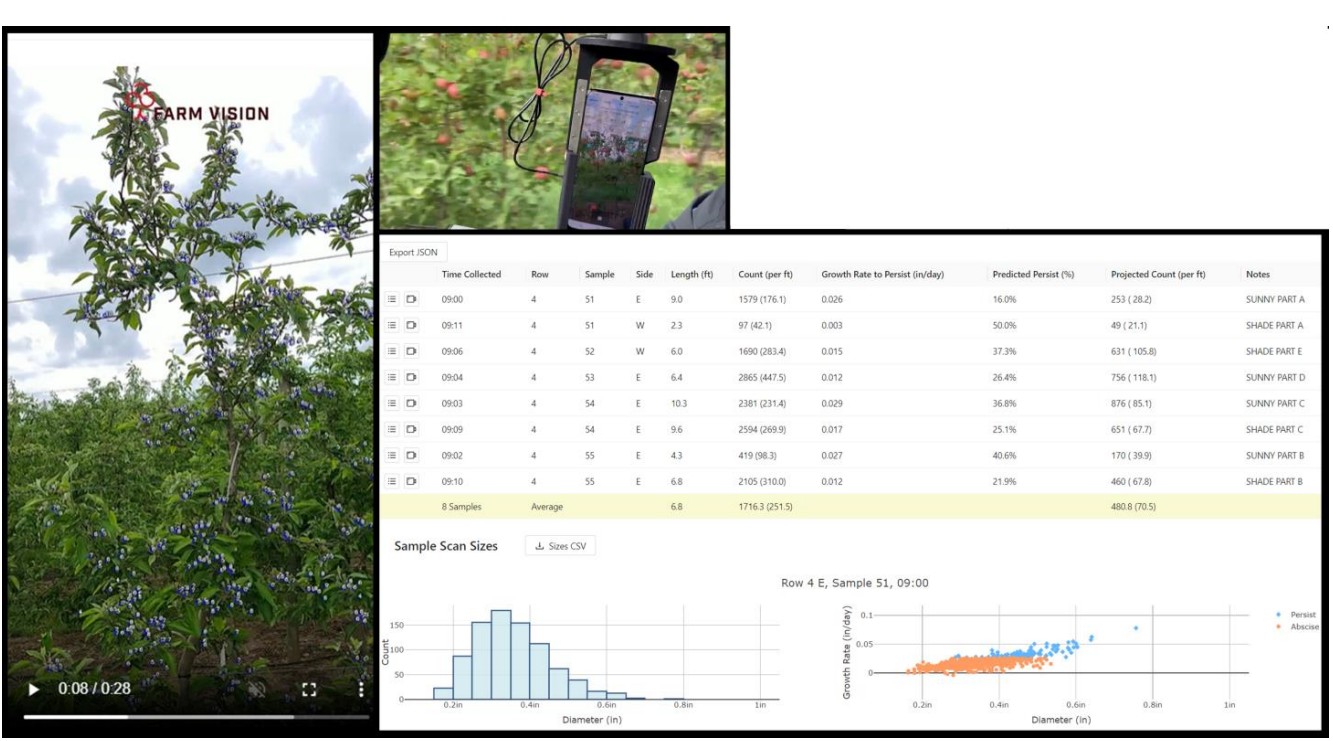

**Figure 6.** Fruitlets detected with the Pometa vision system and information derived from the scans. Blue dots on the tree in the photograph indicate detected fruitlets. The dashboard shows the details of each scan, including number of fruitlets detected, population distribution, and the adjusted number of fruitlets expected to persist (blue dots in graph) or abscise (orange).

**Table 3.** Characteristics of the orchards used to evaluate the fruit growth rate model (FGRM) in 2022.

| Location | Cultivar | Rootstock | System | Spacing (m) |
|---|---|---|---|---|
| Massachusetts | 'Gala' | M.9 | Tall Spindle | 0.9 × 3.7 |
| | 'Fuji' | M.9 | Tall Spindle | 0.9 × 3.7 |
| | 'Honeycrisp' | G.11 | Tall Spindle | 0.9 × 3.7 |
| | 'Gala' | G.41 | Tall Spindle | 0.9 × 3.7 |
| | 'Honeycrisp' | G.41 | Tall Spindle | 0.9 × 3.7 |
| Michigan | 'Honeycrisp' | M.9 | Super Spindle | 0.6 × 3.4 |
| | 'Gala' | G.11 | Super Spindle | 0.6 × 3.4 |
| | 'Fuji' | M.9 | Vertical Axe | 1.5 × 3.7 |
| | 'Gala' | M.9 | Tall Spindle | 1.2 × 3.7 |
| New York | 'Honeycrisp' | B.9 | Tall Spindle | 1.5 × 3.5 |
| | 'Gala' | G.11 | Tall Spindle | 0.9 × 3.4 |
| | 'Honeycrisp' | M.9 | Tall Spindle | 0.9 × 3.4 |
| North Carolina | 'Gala' | M.9 | Tall Spindle | 0.9 × 4 |

## 2.5. Yield Estimation

Yield estimation trials were carried out in 11 commercial orchards in Michigan, New York, and North Carolina in 2021 and 2022. 'Evercrisp', 'Granny Smith', 'Gala', 'Fuji', and 'Honeycrisp' cultivars were evaluated with the Pometa system of yield estimation using iPhone scans and then compared with manual counts of fruit numbers at harvest (Table 4). At each orchard, five uniform trees were selected for yield estimation with the Pometa system, and then manually counted at harvest. Two videos were taken for each tree evaluation (one video for each side of each tree, with a total of ten videos). All videos were uploaded to the Pometa website for processing of the video data. After 24 h, the results of the processed videos were available on the company's website (Figure 7).

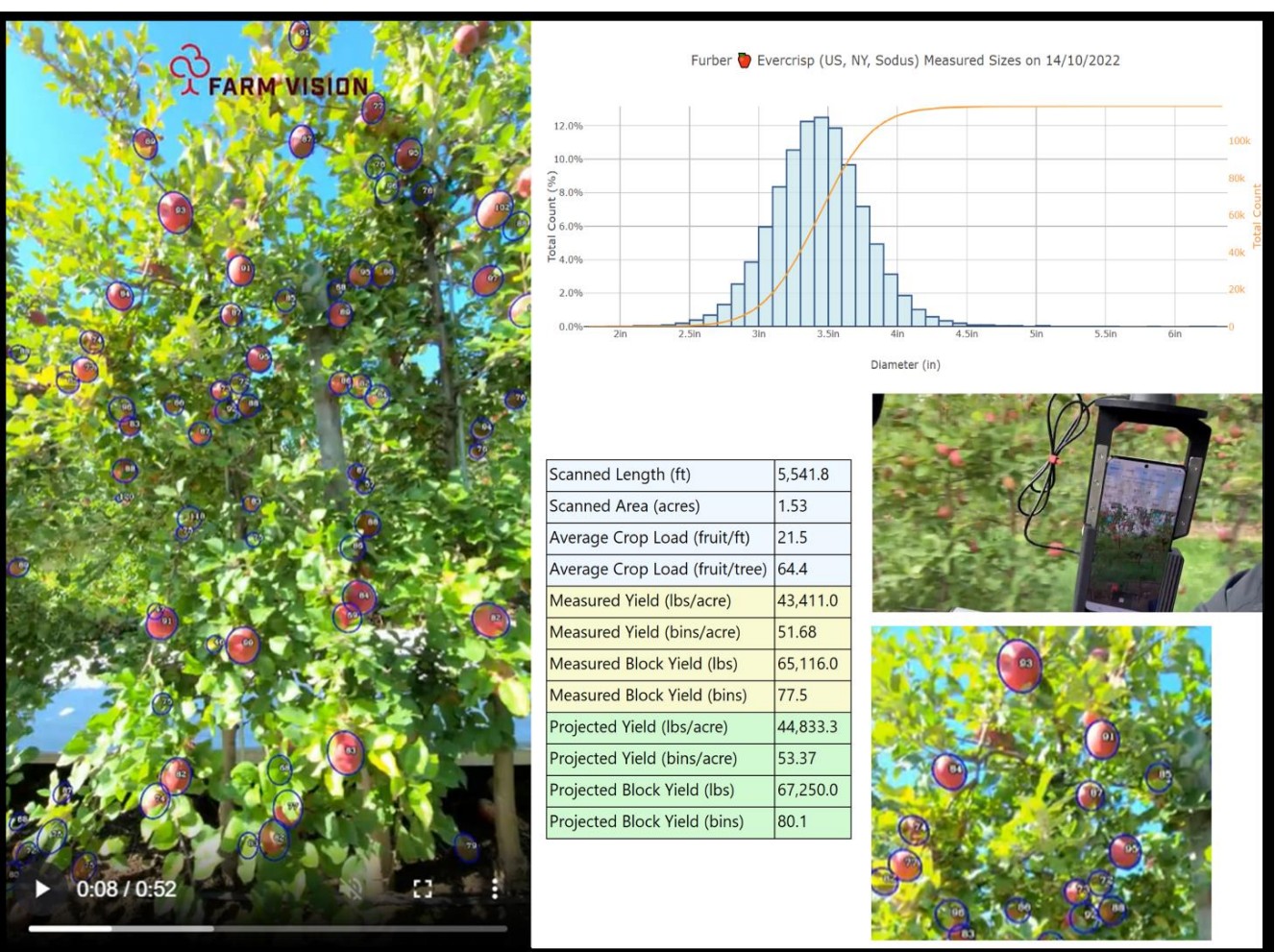

**Figure 7.** Fruit detected with the Pometa vision system and information derived from the scans. Blue circles are fruit detected.

## 2.6. Statistical Analysis

Data were analyzed by linear regression analysis using SAS 9.4 (SAS Institute Inc., Cary, NC, USA, 2009) to evaluate the accuracy of the vision systems in estimating TCSA, bloom intensity, and thinning efficacy, compared with manual measurements. Quadratic regression was used to correlate yields estimation with vision system estimates and manual fruit counts at harvest. The bloom intensity statistical analysis was performed using ANOVA in SAS 9.4 (SAS Institute Inc., 2009, Cary, NC (27513), USA). Means were separated using Fisher's LSD tests at $p < 0.05$.

**Table 4.** Characteristics of the orchards used to evaluate yield estimation technology in 2021 and 2022.

| Location | Cultivar | Rootstock | System | Spacing (m) |
|---|---|---|---|---|
| Michigan | 'Honeycrisp' | M.9 | Super Spindle | 0.6 × 3.4 |
| | 'Gala' | G.11 | Super Spindle | 0.6 × 3.4 |
| | 'Fuji' | M.9 | Vertical Axe | 1.5 × 3.7 |
| | 'Gala' | M.9 | Tall Spindle | 1.2 × 3.7 |
| New York | 'Evercrisp' | B.9 | Tall Spindle | 0.9 × 3.6 |
| | 'Fuji' | B.9 | Tall Spindle | 0.6 × 3.4 |
| | 'Gala' | G.11 | Tall Spindle | 0.9 × 3.4 |
| | 'Honeycrisp' | M.9 | Tall Spindle | 0.9 × 3.4 |
| North Carolina | 'Gala' | M.26 | Vertical Axis | 1.8 × 4.3 |
| | 'Honeycrisp' | M.9 | Multi-leader | 1.8 × 4.3 |
| | 'Granny Smith' | B.9 | Tall Spindle | 0.9 × 3.7 |

## 3. Results

### 3.1. Trunk Cross-Sectional Area

A positive linear correlation was observed between the digital camera measurements of TCSA by the Orchard Robotics system and the manual measurements (Figure 8). The vision system underestimated the TCSA in comparison with the manual measurements in all four trials. The vision system calculated the TCSA using the average of all the visible trunk while the manual measurements were always measured at 30 cm above the graft union.

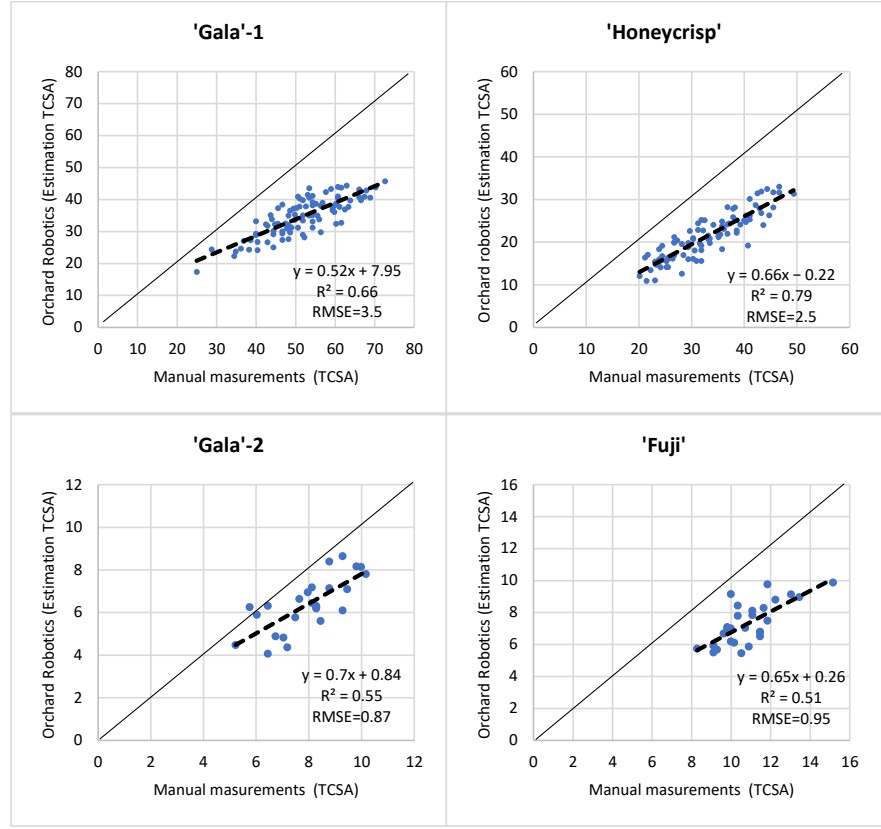

**Figure 8.** Relationships between trunk cross-sectional area (cm$^2$) evaluated by the Orchard Robotics vision system and manual measurements in 2023. 'Gala'-1 and 'Honeycrisp' were older trees (17 years old), and 'Gala'-2 and Fuji were young trees (4 years old). Each symbol represents 1 tree. All linear correlations were significant at *p* < 0.001.

### 3.2. Blossom Counts

Both vision systems evaluated showed a positive linear correlation between estimated number of flowers per tree, measured with the vision systems and manual counts (Figure 9). Overall, both vision systems were accurate with their estimation of flower clusters per tree (1:1), except for 'NY1', in which Pometa underestimated the number of blossoms per tree (Figure 9).

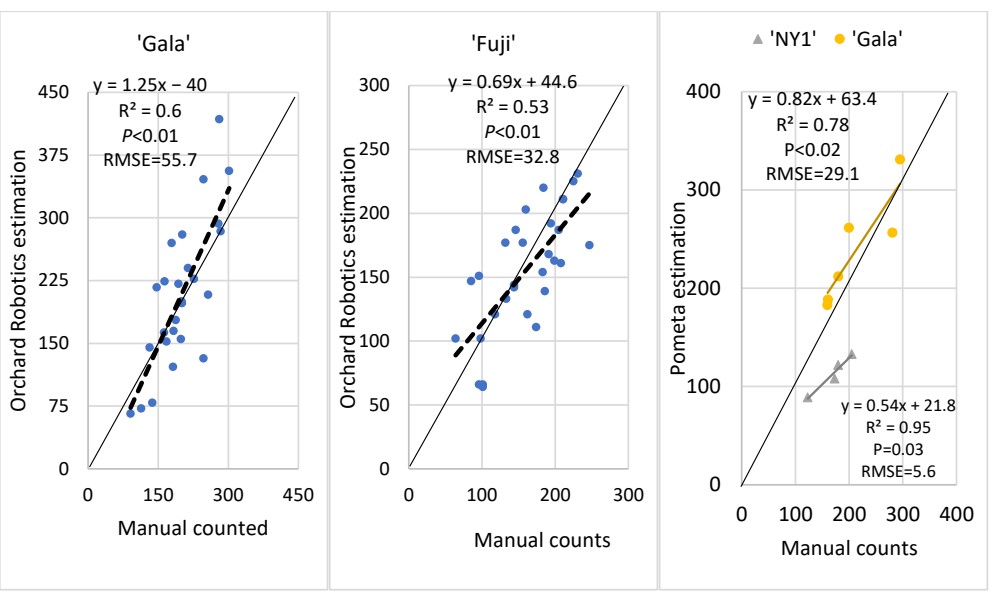

**Figure 9.** Relationships between estimated flower clusters per tree evaluated with the Orchard Robotics and Pometa visions system and manual counts in 2023. Each symbol represents 1 tree.

When all the trees for each cultivar were averaged, the numbers of flower clusters, whether estimated by computer vision of counted manually, were very similar except for the cultivar 'NY1' (Figure 10). However, the regression analysis showed that the manually counted number of flower clusters of 'NY1' was related to the computer vision estimate, but not in a 1:1 relationship (Figure 9).

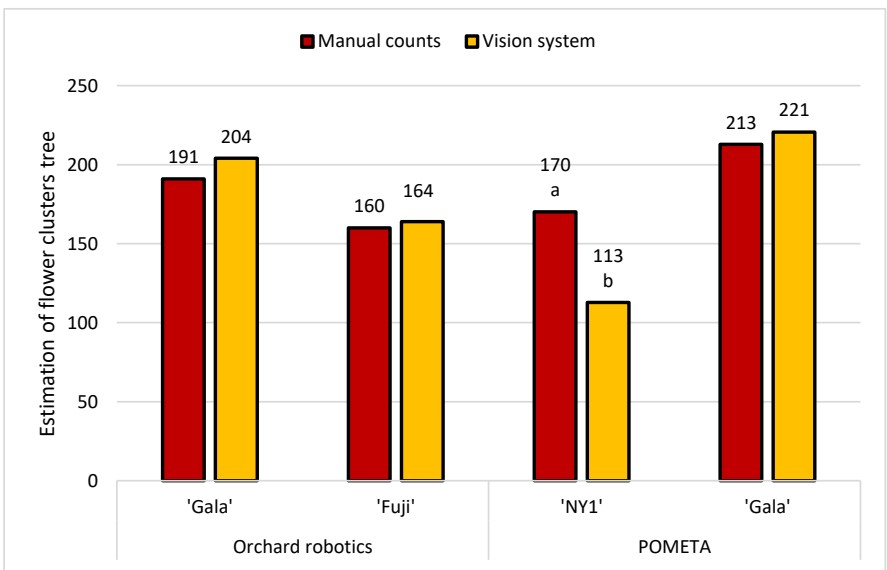

**Figure 10.** Average flower clusters per tree of 'Gala', 'Fuji', and 'NY1', estimated with the computer vision systems of Orchard Robotics and Pometa compared with manual counts in the same orchards (Figure 9). Different letters denote significant differences (Fisher's LSD tests at $p < 0.05$).

*3.3. Thinning Efficacy*

There were significant positive linear relationships between predicted fruit per tree early in the season (June, post-thinning) and final fruit harvested (Figure 11). Both methods (MaluSim and Pometa) overestimated the number of fruit at harvest over a range of different crop loads. When the final numbers of fruit at harvest were lower than 200 fruit/tree, the prediction of both systems was more accurate (near 1:1). That is, the numbers of fruit at harvest were similar to those predicted by each model 7 days after spraying the chemical thinning product. However, when the numbers of fruit at harvest were higher than 200 fruit/tree, both systems overestimated the numbers of fruit at harvest (Figure 11).

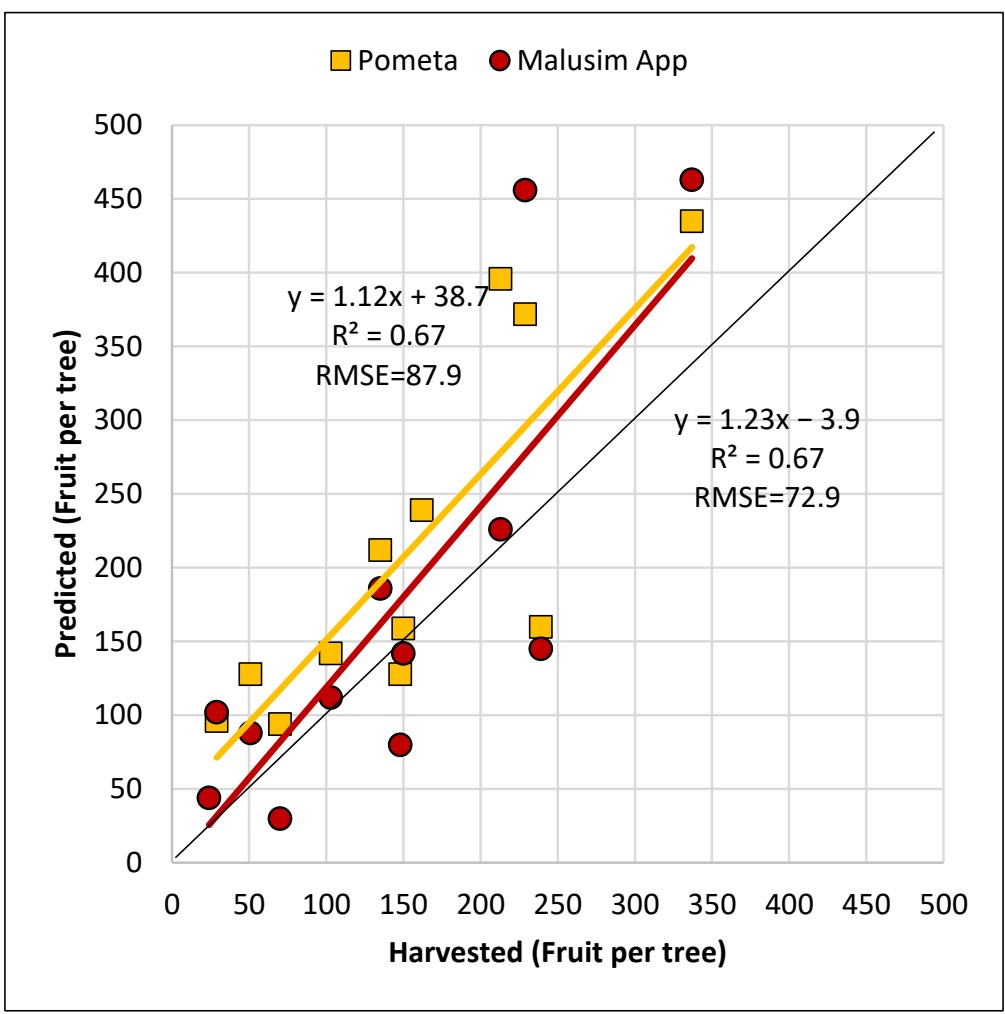

**Figure 11.** Relationships between the actual fruit numbers per tree at harvest in September 2022 and the predicted fruit numbers per tree, using the growth model with the MaluSim app or the Pometa computer vision system 7 days after applying a chemical thinning spray in May 2022. Each symbol represents 1 orchard. All linear correlations were significant at $p < 0.01$.

The predictions of final fruit numbers per tree of both models were correlated (Figure 12). However, when the numbers of estimated fruit were lower than 200 fruit/tree, the Pometa system overestimated more than the MaluSim app. On the other hand, when the numbers of predicted fruit were higher than 200 fruit/tree, both systems showed the same prediction of numbers of fruit.

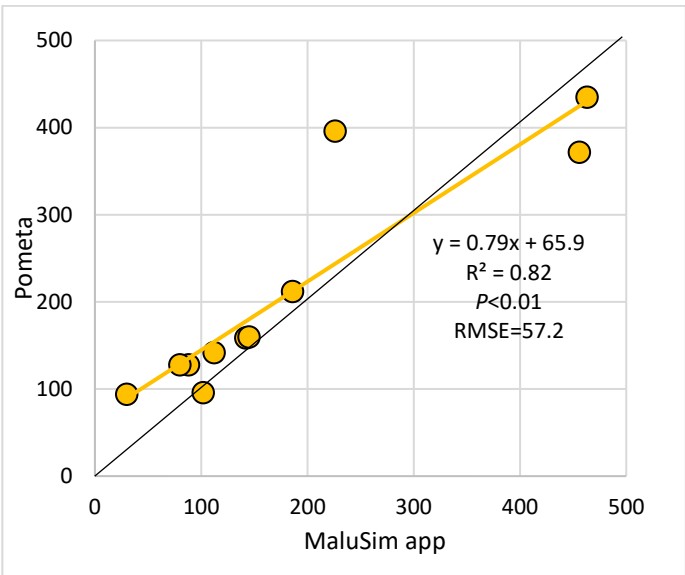

**Figure 12.** Relationship between predicted numbers of fruit per tree with the MaluSim app and the Pometa system in 2022. Each data point represents 1 orchard.

*3.4. Yield Estimation*

There was a significant correlation between fruit yield estimation late in the season (August/September, immediately before harvest) with the Pometa vision system and final yield at harvest (Figure 13). The estimation was accurate (1:1) when the final fruit numbers were fewer than 75 fruit per tree. When the numbers of fruit at harvest were higher than 75 fruit per tree, the vision system underestimated the final yield, except in one plot that overestimated the yield. Moreover, as the final fruit numbers per tree increased, the error in the estimate was higher (Figure 13).

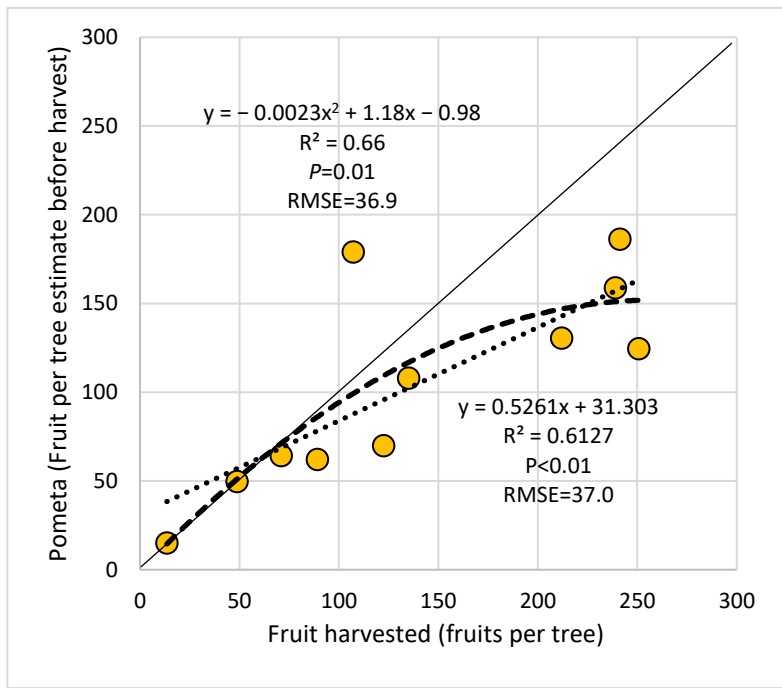

**Figure 13.** Relationship between estimated fruit per tree with the Pometa vision system and actual fruit harvested in 2022. The dotted line represent a linear correlation and the script line is a quadratic regression. Each symbol represents 1 orchard.

## 4. Discussion

Precision agriculture uses information obtained from temporal data to address variability within an orchard, with the objective of reducing the variability through variable management and increase the crop value [23]. As growers adopt digital technologies, it is critical to validate the precision of the new technologies [1,24].

The variation in tree size within an orchard makes establishing a uniform target number of fruit per tree difficult, since the target for each tree is different. TCSA is a metric that can be used to quantify this variability and can be used to estimate optimum tree crop load in terms of wood mass [11,25,26]. Computer vision systems to measure TSCA allow estimates of each tree's optimum economic crop load. Our results show significant variation in TCSA (tree size) around the mean of 40 $cm^2$ in older trees and 4 $cm^2$ in young trees, similar to that shown by Serra, et al. [27]. Our results show that a positive linear correlation was observed between the vision system estimates of TCSA and the manual measurements, similar to the results of Wang et al. [11]. However, in our data, the vision system underestimated the TCSA in comparison with the manual measurements. Wang et al. [11] justified the differences between evaluations with a vision system and manual measurements because the camera sometimes failed to detect the target tree or because there were significant occlusions caused by leaves or overlaps between the foreground and background trunks, concurring with our results. Another reason for the differences between the manual measurements and the vision system is that the vision system calculated the TCSA using the average diameter of all the visible trunk and the manual measurements were always measured at 30 cm above the graft union. The value of using the TCSA as a measure of a tree's capacity to carry fruit is likely very good when the trees are young [25]; however, when trees fully fill their allotted space and arrive at the period of maximum yield, around 10 years after plantation, the TCSA continues to increase, but the yield is constant [28]. Thus, the optimum crop load based on TCSA need to be adjusted down each year as the TCSA continues to increase but the productive capacity of the tree remains constant. With the proper adjustment, the TCSA could still quantify the variability in tree size in an orchard to allow differential crop loads based on the trees' capacity.

Both of the vision systems showed a positive linear correlation between numbers of flower clusters at full bloom and manual counts. This is consistent with the results of two other systems, the Cartographer [29] and unmanned aerial vehicles [30]. The Orchard Robotics system consistently provided accurate counts when properly calibrated with manual counts to account for occluded flower clusters. The Pometa system showed variability in the results, with one block showing accurate counts and another with large differences between the manual flowers counted and those detected with the vision system. However, both blocks showed a high linear correlation, and it is possible that the overall error could be reduced with a correction factor based on manual counts of representative trees. It appears that both systems could be used to manage the flower thinning with precision. For example, Penzel, et al. [31] considered estimates of flower intensity as essential to manage mechanical flower thinning. In addition, these systems could be used to estimate the best timing to initiate the pollen tube growth model, developed by Yoder, Peck, Combs and Byers [16], and Peck, Combs, DeLong and Yoder [17]. The geo-referenced information on flower cluster density can be used to create accurate heat maps to guide variable smart sprayers to manage the variable rate application of chemical thinning agents to match chemical dosage with flower cluster density on each tree.

Greene et al. [19] developed a model (fruit growth rate model) to predict the efficiency of thinning and to inform decisions on the necessity of more thinning sprays. The model uses the growth rate of tagged and measured fruitlets to predict the percentage that will abscise. A mobile phone app (MaluSim) was developed by Francescato and Robinson (unpublished) to automate the calculations necessary to run this model, which improved the efficiency and accuracy of this method, although it still requires manual measurements of fruit diameters. The recommended method is to measure five representative trees × 15 clusters per tree × 5 fruitlets per cluster, for a total of 375 fruitlets. Manual

measurements require about two hours of work per orchard on each date. At least two measurements are required to estimate fruit per tree. The Pometa vision system uses the same concept of the fruit growth rate model but uses computer vision to measure the fruit diameters. This system utilizes a cell phone camera to capture videos of five representative trees. The videos are processed by a proprietary algorithm to determine which fruitlets will abscise. The time required to carry out all measurements is about 10 min per block. Ours results suggest that both systems (MaluSim and Pometa) showed accurate predictions of the numbers of fruit at harvest when the final fruit numbers were lower than 200 fruit/tree, concurring with Penzel, et al. [32], Costa, et al. [33], and McArtney, et al. [34]. However, our results suggest that, when the numbers of fruit at harvest were higher than 200 fruit/tree, both the MaluSim and the Pometa systems overestimated the numbers of fruit at harvest, which concurs with the results of Rufato, et al. [35]. The overestimation of fruit numbers per tree is likely due to the higher natural fruit drop after the chemical thinning window has passed [36]. This is typically attributed to a carbohydrate deficit in June, resulting in "June drop". Our results also suggest that, when the numbers of fruit are higher, they consume greater amounts of carbohydrates and, as a consequence, the trees experience more frequent carbohydrate deficits, resulting in higher natural fruit abscission than in trees with a lower fruit load. Despite the overestimation of final fruit numbers when the crop load is high, our results show that both the manual model (MaluSim) and the computer vision system of Pometa are correlated, and the estimates with the computer vison system are very close to the estimates with manual measurements. Thus, the vision system with a cell phone camera will likely become a preferred method to estimate thinning efficacy in the future since it requires less time and provides easy interpretation of the results.

Estimation of harvest provides important information to apple orchard managers, storage operators, and fruit marketers. For automated harvests, yield maps will be required. Apple yield estimation with a smartphone offers the advantages of low cost, quick access, and simple operation [37]. Our results suggest that the use of smartphones to carry out yield estimation produced accurate results (1:1), until the numbers of fruit exceeded 75 fruit per tree. Similar results were shown by Wang, et al. [38] with a Nikon D300s camera. However, when the numbers of fruit at harvest were higher than 75 fruit per tree, the Pometa vision system underestimated the final yield because there was significant fruit occlusion caused by foliage. Scalisi, et al. [29] showed good correlations between predictions with the Cartographer system and manual measurements as long as the maximum fruit number per tree was 80 fruit, which is similar to our results. Penzel, et al. [39], working with Lidar, showed better predictions with a high range of numbers of fruit per tree. Currently, the Pometa company is working on an occlusion model to improve the yield estimation with tree forms that have higher occlusion of fruit.

## 5. Conclusions

The use of the vision systems to estimate TCSA, bloom intensity, thinning efficiency, and yield generated accurate predictions and usable data for the growers. The vision systems could measure the TSCA for each tree to estimate the optimum fruit numbers for each tree. However, the vision system underestimated the TCSA in comparison with the manual measurements because the measurements with the vision system were carried out in a different place on the trunk. Both vision systems evaluated could estimate the numbers of flower clusters per tree accurately. This information when geo-referenced with each tree could allow precise management of flower-thinning spray applications. The fruit growth rate model using the MaluSim app and the Pometa system showed accurate predictions of thinning efficacy when the crop load was lower than 200 fruit/tree. However, our results suggest that, when the numbers of fruit at harvest were higher than 200 fruit/tree, both the MaluSim and the Pometa systems predicted higher numbers of fruit at harvest than were actually harvested. The vision system of Pometa with cell phones needed less time than the MaluSim method and provided easy interpretation of the results. The yield estimation with Pometa was accurate up to a crop load of 75 fruit per tree. However, when the numbers of

fruit at harvest were higher than 75 fruit per tree, the Pometa vision system underestimated the final yield because there was significant fruit occlusion caused by foliage. Currently, Pometa is working with a new occlusion model to improve the yield estimation with tree forms with high occlusion of fruit.

Overall, precision crop load management with smartphone-bases systems offers advantages such as low cost, quick access, and simple operation. However, the vision systems with special cameras and geo-referencing of each tree offer tree-specific information that will allow precise crop load management of each tree.

**Author Contributions:** Conceptualization, L.G.N., T.L.R., A.W., J.C., M.M.S., C.K. and T.M.K.; methodology, L.G.N., T.L.R., A.W., J.C., M.M.S., C.K. and T.M.K.; software, L.G.N.; validation, L.G.N., T.L.R., A.W., J.C., M.M.S., C.K. and T.M.K.; formal analysis, L.G.N., A.W. and J.C.; investigation, L.G.N., T.L.R., A.W., J.C., M.M.S., C.K. and T.M.K.; resources, L.G.N., T.L.R., A.W., J.C., M.M.S., C.K. and T.M.K.; data curation, L.G.N., T.L.R., A.W., J.C., M.M.S., C.K. and T.M.K.; writing—original draft preparation, L.G.N.; writing—review and editing, L.G.N., T.L.R., A.W., C.K. and T.M.K.; visualization, L.G.N., T.L.R., A.W., J.C., M.M.S., C.K. and T.M.K.; supervision, T.L.R.; project administration T.L.R.; funding acquisition, T.L.R.; All authors have read and agreed to the published version of the manuscript.

**Funding:** This research was funded by the Specialty Crop Research Initiative project "Precision Crop Load Management of Apples" (NYG-632521).

**Data Availability Statement:** Not applicable.

**Acknowledgments:** This work was supported by the USDA National Institute of Food and Agriculture—Specialty Crop Research Initiative project "Precision Crop Load Management of Apples" (NYG-632521). We thank the Orchard Robotic (https://www.orchard-robotics.com) and Pometa (https://pometa.io) companies.

**Conflicts of Interest:** The authors declare no conflict of interest.

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
