# Peer review of "Evaluation of Computer Vision Systems and Applications to Estimate Trunk Cross-Sectional Area, Flower Cluster Number, Thinning Efficacy and Yield of Apple"

_horticulturae, doi:10.3390/horticulturae9080880_

Round 1

Reviewer 1 Report

Computer vision systems that can detect fruit, flowers and measure the trunk cross sectional area of fruit trees are highly valuable for future tree adapted crop load management.

The strength of the study is that, a commercial product already exists for this purpose and it can record these features of the trees in a short time. In the study, the precision with which the data are captured by different systems was tested in different orchards with different varieties and plant spacing.

The contribution of the study in comparison to other studies who focused on vision systems needs to be more described and explained. So far there are not so may commercial solutions for that purpose available but many studies with probably days or weeks of manual data analysis are existing. For me a more detailed comparison between existing approaches and the presented approach was missing .

Specific comments:

Abstract:

-can be improved

-you can already provide R² of the correlations which you have found between manual and computer vision measurement

Introduction

-There are several more studies on counting flowers, fruits and recording TCSA with different kinds of optical sensors, maybe you can add some more references

- Please avoid the royal we/us. It can be assumed that the experiment and results are reproducible. Consequently, results are objective.

- maybe add a hypothesis

 Keywords:

-please avoid to name commercial products as keywords

Materials and Methods

-add the number of trees for each orchard which were used for the trial

-please add a flow chart of the steps for data processing. Then it would be possible to compare the systems-how many leaves were sampled?

-it may improves the understanding which measurement was done in the individual orchards by grouping information of table 1-4 into one table.

-Line 107: Until which height has the robotic vision system scanned the trunk? Was the part at 30 cm above the graft union always within the scanned part?

-check line 148, iphone is a commercial product, maybe its better to call it cell phone scans? Or is that app only available to i phones? Same in line 186

-line 152: isnt it called "king fruit"?

Results, Discussion

-you dont need to mention p values if you have defined in M&M what significant means in your study. Then in the results part R² are enough. E.g. in figure 9 or in line 209

-I havent understand if figure 10 contains fruit from different orchards or only fruit from the same orchrd? Can you specify that? Were there differences between orchards?

-was there a correlation between tcsa and fruit/yield per tree in the orchards in general? That correlation has been shown in several studies with young trees but there are also studies who havent seen a correlation because trees were old

-what is your conclusion tot he underestimation of TCSA? IS it still possible to calculate precise target fruit numbers per tree from the machine vision scanned TCSA? Would further calibration help?

Author Response

Hi  
Thanks for your review. 
I accept practically all suggestions.
 I answer in color red 
Thanks

Reviewer 2 Report

The aim of the current study was to evaluate and compare different commercial computer vision systems and computer applications to estimate trunk cross-sectional area (TCSA), flower cluster number, thinning efficacy, and yield estimation. We evaluated the digital methods of two companies that offer different vision systems in a  series of trials across orchards in four states.

(1)The abstract should be improved. Your point is your own work that should be further highlighted.

(2)The parameters in expressions are given and explained.

(3) The method in the context of the proposed work should be written in detail

(4) The values of parameters could be a complicated problem itself, how the authors give the values of parameters in the used methods.

(5) The literature review is poor in this paper. You must review all significant similar works that have been done. I hope that the authors can add some new references in order to improve the reviews and the connection with the literatures. https://doi.org/10.3389/fendo.2022.1057089; https://doi.org/10.1016/j.ins.2023.03.142; http://dx.doi.org/10.1016/j.marstruc.2022.103338; http://dx.doi.org/10.1145/3513263 and so on.

(6) In Section 2.4, at Line 157, why is 75 clusters in total?

(7) How to set Parameter values of the used methods?

Extensive editing of English language required

Author Response

(The authors gave the same response as above.)

Reviewer 3 Report

Dear Authors,

The subject of the study is interesting and topical, with scientific and practical importance.

The introduction is presented correctly, in accordance with the subject. Numerous scientific articles, in concordance to the topic of the study, were consulted.

Methodology of the study was clearly presented, and appropriate to the proposed objectives.

The obtained results are important and have been analyzed and interpreted correctly, in accordance with the current methodology.

The discussions are appropriate, in the context of the results, and was conducted compared to other studies in the field.

The scientific literature, to which the reporting was made, is recent and representative in the field.

Some suggestions and corrections were made in the article.

The following aspects are brought to the attention of the authors.

1.

Malus domestica Borkh.” instead of “Malus domestica Borkh.”

Italic Font style for species name

2.

“cm2

“R2’ ”

3.

Different ways to write the names of apple cultivars

e.g.

Page 2, 2.1. Plant material and sites, ’Gala’,

Page 6, Table 3, Gala

Similarly for the other apple cultivars

Page 2, 2.1. Plant material and sites, ‘NY1’

Page 4, Table 2, ‘NY-1’

It is recommended to review the entire article and the unitary presentation of the names of the apple cultivars studied.

4.

Citation of bibliographical sources in the text

Please revise the citation of some bibliographical sources in the text, according to Instructions for Authors, and Microsoft Word template, Horticulturae journal.

Some suggestions were made in the article

5.

References

According to Instructions for Authors and Microsoft Word template, Horticulturae journal,

Author 1, A.B.; Author 2, C.D. Title of the article. Abbreviated Journal Name Year, Volume, page range.

Include the digital object identifier (DOI) for all references where available.

e.g.

“Gonzalez, L.; Torres, E.; Àvila, G.; Bonany, J.; Alegre, S.; Carbó, J.; Martín, B.; Recasens, I.; Asin, L. Evaluation of chemical fruit thinning efficiency using Brevis® (Metamitron) on apple trees (‘Gala’) under Spanish conditions. Sci. Hortic. 2020, 261, 109003. doi:10.1016/j.scienta.2019.109003”

Instead of:

“Gonzalez, L., et al., Evaluation of chemical fruit thinning efficiency using Brevis® (Metamitron) on apple trees (‘Gala’) under Spanish conditions. Scientia Horticulturae, 2020. 261: p. 109003.”

Please check the entire References chapter and correct as necessary

Author Response

(The authors gave the same response as above.)

Reviewer 4 Report

Evaluations of digital technologies for estimation of trunk cross-sectional area, flower cluster number, fruit set, and yield of apple

Dear Authors

The basic science of this paper is conducted in a good way and is of an appropriate standard.  The author and his team write this paper according to the journal's scope and modern trends. I already published several papers in this domain. I am glad to review this paper because this manuscript is very relevant according to my research.  The current study was to evaluate and compare different commercial computer vision systems and computer applications to estimate trunk cross-sectional area (TCSA), flower cluster number, thinning efficacy, and yield estimation. We evaluated the digital methods of two companies that offer different vision systems in 20 series of trials across orchards in four states. Moreover, the author needs more time to modify this paper. Figures are appropriate according to journal standards. Some important comments are embedded below

Major comments

·         Title is not appropriate.

·         Abstract is inappropriate and there is no significance according to international journal criteria. I can’t find the objectives, methodology, and results of both datasets

·         Moreover, the abstract section does not reflect the whole research.

·         Rewrite the whole abstract section because the abstract section does not reflect the whole study. Moreover, the abstract section is very complex and there is no continuity in the sentences. Why the author used this study, the author should focus on the main aim of the research

·         Remove some keywords and keywords should be 5.

·         Introduction section is very lengthy and there is no sequence of this research. Many statements are out of justification.

·         Recheck the language of the whole manuscript.

·         Introduction section is not appropriate and the problem statement, research questions, and hypothesis are missing from the introduction section.

·         Introduction section is very simple and I did not find the research question, problem statement, and innovative idea of this research.

·         Figure 1 is not according to journal criteria. The author should prepare according to journal criteria.

·         Rest of everything is fine in this manuscript.

Best Regards

Author Response

(The authors gave the same response as above.)

Reviewer 5 Report

The presented study is an interesting comparison between two commercial solutions for automating useful information extraction for apple orchard management. The overall paper quality is high, materials and methods, as well as results are well presented, and discussion properly highlight pros and cons in each solution investigated.

Some suggestion to improve the manuscript are reported below.

Typos / text error :

- lines 119-120: re-word for readability the sentence "Pometa cluster counting in 2023 beta product for the company and is not yet offered as a commercial product". I understand but is not well written.

-line 224: change 'lineal' in 'linear'

-line 320: correct the citation of Yoder et al.; there is an open (and never closed) parenthesis and the number 2 before the author

if possible, add somewhere links to websites of orchard robotic and Pometa companies (maybe in the acknoledgements).

Images and table seem to not be well placed (i.e., centered) with the text, but this could be due to the not-final version of the paper. chekc it.

Screenshot from the websites of the presented solutions could be improved for their quality (if possible - quality of fig. 5 is ok!) since readability is difficult for presented parameters in the pictures. These pictures might be added in bigger size as 'adding materials'. 

Results:

-clear presentation, but if possible, I suggest to include other metrics than regression in order to better highlight the accuracy of the systems compared with the manual measurements (e.g., average error or RMSE or other metrics preferred by the authors)

-in fig.13 could be presented also the linear regression line ( and its R2 value) to be consistent with all the other presented regression.

-in fig 11 regression lines could be of the same color of the data to which they refer.

Author Response

Hi  
Thanks for your review. 
I accept practically all suggestions.
 I answer in color blue
Thanks

Round 2

Reviewer 4 Report

The author should modify figure according to international journal criteria. add geolocation [latitude and longitude] in figure 1.